# Screen Exposure in 4-Year-Old Children: Association with Development, Daily Habits, and Ultra-Processed Food Consumption

**DOI:** 10.3390/ijerph21111504

**Published:** 2024-11-13

**Authors:** Gabriela M. D. Gomes, Rafaela C. V. Souza, Tamires N. Santos, Luana C. Santos

**Affiliations:** Escola de Enfermagem, Universidade Federal de Minas Gerais, Belo Horizonte 30130-100, MG, Brazil; gabidias2012@gmail.com (G.M.D.G.); tamiresndossantos@gmail.com (T.N.S.); luanacstos@gmail.com (L.C.S.)

**Keywords:** screen exposure, screen time, daily screen time, child development, food behavior, ultra-processed foods

## Abstract

This study aimed to investigate the association between child development, daily habits, and ultra-processed food consumption with screen exposure in 4-year-old children. A cross-sectional study was conducted using a questionnaire that included sociodemographic data, the child’s daily habits, and screenings for child development and eating habits. The daily screen exposure time (cell phone, computer, television, and/or tablet) was measured in minutes and classified as inadequate if >60 min. We conducted bivariate analyses and a generalized linear model. Overall, 362 caregivers–children pairs were investigated. The average screen time per child was 120 min (IQR: 120), and most of the children (71%) showed inadequate screen time for the age group. The longest screen time was associated with the lowest score in child development (β = −0.03; *p* = 0.01), an increased habit of eating in front of screens (β = 0.34; *p* < 0.001), and the highest score of ultra-processed foods (UPFs) (β = 0.05; *p* = 0.001). The sample showed a high prevalence of inadequate screen time, and this has been associated with the lowest score in child development, an increased habit of eating in front of screens, and the highest score of UPFs.

## 1. Introduction

Excessive screen exposure negatively impacts many health indicators in infants, such as motor, cognitive, and socioemotional development, sleep quality, language development, and literacy [1,2,3]. The literature shows that the longest screen time for children and teenagers was associated with the increase in adiposity, besides being connected with higher calorie intake and lower nutritional diet quality [4]. Additionally, screen exposure allows access to marketing and food publicity, which directly impacts food preferences and behavior. Also, they are mostly about ultra-processed foods (UPFs) in the channels on free-to-air Brazilian television [5,6].

Some factors seem to impact the screen time in children, for example, the lowest income and the lower level of education of the caregiver, which can lead to a higher screen time [7,8].

Therefore, the World Health Organization published in 2019 some guidelines for physical activities, sedentary behavior, and sleep for children under 60 months [9]. In these guidelines, non-interactive screen time—passive screen exposure time, in which one does not have to move or exercise—for 2 to 4-year-old children was considered adequate if lower than 60 min a day.

Despite this recommendation, epidemiological data show a high prevalence of inadequate screen time in preschool children. A national study conducted in 2022 showed that 33.2% of children under 59 months watch TV programs or play games on TV, smartphones, or tablets for over 120 min a day [10]. A relevant report on the American population showed a similar scenario, showing an average of 150 min a day in children aged 24 to 48 months old [8].

The researchers that investigated screen exposure time and its association with health indicators in children are mostly from countries in North America and Europe [11]. There are few Brazilian studies conducted showing the relation between screen exposure time with the screening of child development, and the screen exposure time was associated with ultra-processed food consumption using a validated instrument [12,13]. Among the main hypotheses of this work, it is worth highlighting that delays in child development and ultra-processed food consumption in early childhood could be associated with excessive exposure to screens.

Therefore, the aim of this study was to investigate the association between screen exposure and child development, eating habits, especially ultra-processed foods, and daily habits with 4-year-old children.

## 2. Materials and Methods

### 2.1. Study Design, Sample, and Data Collection

This is a cross-sectional study of the last moment of a prospective cohort in which puerperal women and their babies were initially followed after the immediate postpartum between March 2018 and October 2019 in two public hospitals of Belo Horizonte and then at 6 months, 12 months, and 4 years old with the presence of the main caregiver. The exclusion criteria adopted during postpartum were age under 18 years, twin pregnancies, prepregnancy diabetes, AIDS, and complications during pregnancy, including severe hypertension (eclampsia and preeclampsia) and gestational diabetes [14].

A pilot study was conducted to test and validate the instruments used. Then, data were collected between March 2022 and October 2023. Nutritional and scientific initiation students, previously trained, collected the data remotely on the date and time scheduled.

Initially, the families were invited to take part in the research via a phone call with the main caregiver around a month before the child turned 4 years old. The invitation aimed at continuing the follow-up started months before. Once the caregiver agreed, we scheduled a teleconsultation through the app WhatsApp Messenger^®^ with a duration of around 1 h. The families that could not be reached through phone calls, either because of a change in number or non-existence, were searched on Facebook^®^ and Instagram^®^.

The sample calculation was based on a previous study [15] and considering a 95% confidence interval, 80% test power, and the presence of 8 predictors in the regression model to predict the factors associated with screen time. Therefore, a 260-participant sample is necessary. The calculation was performed in the software GPower, version 3.1.9.2.

Overall, 432 families participated in the teleconsultation, and those families that did not answer about daily screen time (*n* = 70) were excluded, resulting in 362 caregiver–child pairs, exceeding the minimum estimated number and ensuring the necessary representativeness for data analysis.

### 2.2. Study Variables

During the interview, the caregiver answered questions from a semi-structured questionnaire prepared by the authors, which was composed of questions about themselves and the children. We collected the following data about the caregivers: full name, telephone, full address, relationship with the child, age (in years), self-declared color (white, brown, black, Asian, or Indigenous), marital status (with or without a partner), level of education (finished elementary school, finished high school, or higher education or above), professional occupation (paid or unpaid), use of social benefits (yes or no), number of household members, and total monthly family income. The per capita income was found by dividing the total monthly family income and the number of household members.

The caregiver answered questions about the child: age (in months), sex (female or male), color (white, brown, black, Asian, or Indigenous), level of education (if attended school/childcare center, or not), school hours (full-time or part-time), age that started school (in months), breastfeeding (in months) and if the child had alterations or suspicions of developmental delays (e.g.,: attention deficit hyperactivity disorder, autism spectrum disorder, motor disorders and learning delays) or behavioral problems (yes or no).

To assess the child’s nutritional status, we collected weight and height data from the Child Health Booklet [16], and subsequently, the growth standards were analyzed: weight by age, length by age and body mass index (BMI) by age. The classification of nutritional status was made according to the curves recommended by the World Health Organization [17]. For the weight-for-age index, z-score values ≥−2 and ≤2 were considered “adequate”, while the “inadequate” classification was established for z-score values <−2 or >2. As for the length-for-age index, z-score values ≥−2 were considered “adequate”, while the “inadequate” classification was assigned to z-score values <−2. Finally, the BMI classification was considered “adequate” for z-score values ≥−2 and ≤1, while the “inadequate” classification was established for those who had a z-score <−2 or >1 (Table 1).

In addition, we applied during the interview the screening instrument Survey of Wellbeing of Young Children (SWYC) [18], which was validated for the Brazilian population [19]. That is an instrument composed of 10 questions directed to the main caregiver which identifies delays in child development through questions related to the cognitive, language, and motor domains. For each answer, a score was added on a 3-point scale according to the child’s performance: “not yet” = 0, “somewhat” = 1, and “very much” = 2. At the end of the questionnaire, the child presented a score of 0 to 20 points, considering that, as the higher the score, the lower the suspicion of developmental delay.

Information on the child’s daily habits was also collected, including daily reading practice at home conducted by the responsible person during the previous week (0 to 7 days), average daily sleep of good quality (with regular time to sleep and wake up, including naps—in hours), and if the child had the habit of eating in front of screens (cell phone, computer, television, and/or tablet—yes or no).

Daily screen time (cell phone, computer, television, and/or tablet) was informed from an estimate made by the main caregiver (in minutes) and classified according to the recommendation of the World Health Organization [9].

Additionally, new questions about the children’s eating habits, according to the validated questionnaire for Brazilian population NOVA of ultra-processed foods score (UPF score), were included [12,13]. This system includes 23 subgroups of the most consumed UPFs by the Brazilian population according to a national form of food consumption [19]. These foods are divided into three categories: beverages (*n* = 6), products that replace or are eaten with main meals (*n* = 10), and foods normally eaten as snacks (*n* = 7) and, for each food group consumed the day before, one point is added, reaching from 0 to 23 points. Therefore, as the higher the score, the higher was the consumption of UPFs.

Finally, considering the COVID-19 pandemic, which started in 2020, and its potential short and long-term impact on the outcome, the child’s infection with SARS-CoV-2 was investigated at some moment (yes or no).

### 2.3. Statistical Analysis

The data were tabulated in the software Epicollect5 Data Collection^®^(v5.1.54), and then a consistency analysis was carried out to find possible typing errors. Later, the software IBM Statistical Package for the Social Sciences (SPSS) version 19 performed the statistical analysis, and the significance level *p* < 0.05 was adopted for all the statistical tests conducted.

A priori, the Kolmogorov–Smirnov normality test evaluated the variables’ symmetry. A posteriori, descriptive statistical (absolute and relative frequency), measurement of central tendency (median), and dispersion (interquartile range, IQR) were calculated.

The Mann–Whitney and Kruskal–Wallis tests were applied to compare the average screen time according to the categorical characteristics of the caregiver and the child. Spearman’s correlation test was also applied to evaluate the correlation between screen time, the caregiver’s characteristics (age, number of family members, total income and per capita income), and the child’s characteristics (age, the age that started school, breastfeeding, child development, daily reading, sleep time, and UPF score).

The multivariate analysis was carried out through the generalized linear model with gamma distribution and logarithm function aiming at finding the association between the outcome (screen time) and the explanatory variables. The variables that presented *p* < 0.20 in the bivariate analysis were added to the model, eliminating those that presented lower statistical significance, according to the backward method. In the final version of the model, all the variables presented *p* < 0.05. The variables used as adjustments were those that demonstrated greater relevance for the outcome explanation: age, marital status, level of education, per capita income, and infection of the child with SARS-CoV-2. The values of the final model were expressed in β, 95% confidence interval (CI 95%), and *p*-value. The F-test assessed the significance of the variance analysis of the final model, and the adjustment quality was evaluated by the coefficient of determination (R2).

## 3. Results

A total of 362 families were assessed; the main caregivers were mostly mothers (95.6%), and the children’s average age was 48 months old (IQR: 0).

The average screen time was 120 min (IQR: 120), and most of the children (71%) had inadequate screen time for the age group. Moreover, they were also frequently exposed to screens while eating (64.3%).

The bivariate analysis showed that the children presented the highest screen time when they were part of a household with a low per capita income, did not attend school or a childcare center, had low scores in child development, had few days of daily reading, had exposure to screens while eating, and had higher UPF scores (Table 1).

The multivariate analysis showed that the longest screen exposure time is associated with the lowest score in child development with the increased habit of eating in front of screens and the highest UPF score. Those variables contributed around 35% to explain the outcome even after the adjustments (Table 2).

## 4. Discussion

This study identified a high average (120 min) and high prevalence of inadequate 175 screen exposure time (71%). Such exposure was associated with the lowest score in child development, an increased habit of eating in front of screens, and the highest UPF score in 4-year-old children. These findings support the hypothesis that ultra-processed food consumption and delays in child development in early childhood are associated with excessive exposure to screens.

The exposure of young children to screens has been widely studied, and a similar result (90 min) was observed among 60-month-old children [7]. Other studies that used the same cutoff point that we used in relation to inadequate screen time (>60 min) also presented high prevalence rates [11,20]. A study with 856 Canadian children identified that the average screen time was 120 min in children from 3 to 4 years old and that 78% showed high screen time (>60 min) [20]. In 3155 Brazilian children aged 0 to 60 months, we identified an increase in screen time (>60 min) according to their age, reaching a percentage of 85.2% in children aged 49 and 60 months old [11].

It should be noted that there has been an increase in the screen time of Brazilian children after the COVID-19 pandemic [21] given the influence of the lockdown in the sedentary behaviors in all life cycles [22,23]. An Asian study that analyzed screen time among 630 children aged 3 to 10 years before and during the pandemic showed an increase of 1.2 h (*p* < 0.001) of screen time [24]. An important Polish study that evaluated children and adolescents aged 6 to 15 years in the same period identified a 3.8% increase in the percentage of children who watched television or programs on the internet more than 6 h a day during the week [22].

Among our main findings, we identified an inverse relationship between longer screen time and child development score, which is similar to the results of previous studies that used similar screening instruments [7,11,25]. A Brazilian population-based study showed that each additional hour of screen time for children between 0 and 60 months was able to reduce different domains of child development, such as communication (*p* < 0.001), problem-solving skills (*p* < 0.001) and personal–social domain scores (*p* < 0.001) [11]. In addition, a systematic review demonstrated unfavorable associations between screen time and cognitive development indicators such as language, number recognition, classroom engagement, attention problems, and delayed executive function [2].

Such developments can be explained by the fact that early childhood is a crucial period in child development when the brain has great brain plasticity, and the formation of new neuronal circuits and the maturation of social, cognitive, and emotional domains occur, making these children highly sensitive to environmental stimuli [1,26]. Therefore, screen exposure in this period can compromise the child’s ability to fully develop and lead to negative outcomes in developmental milestones [1,27], as evidenced in the screening applied in this research.

Our study also showed a direct association between screen time and the habit of eating exposed to screens. Data from the Food and Nutrition Surveillance System (SISVAN) showed that of 502,101 Brazilian children aged 24 to 48 months evaluated in 2023, 53% had the habit of eating in front of screens [28]. Another Brazilian study conducted in the first year of the pandemic revealed that 54.3% of children had the habit of eating in front of screens [29]. Among 210 American children aged 12 to 36 months, it was observed that screen exposure during the first meal of the day increases by 83% the tendency of this habit to be repeated during large meals [25]. It is important to note that lifetime habits are established during childhood, for example, eating behaviors. These are susceptible to the influence of the environment, such as screen exposure, food advertising, and the eating habits of parents and caregivers [30].

Given this context, it is suggested that the most consumed foods by the investigated children belong to the ultra-processed group, and several studies indicate that the foods predominantly ingested during screen exposure are those with high energy density [27,31]. Added to this, it is important to consider that UPFs dominate advertising and marketing on screen devices, and children are often the focus considering the target audience. A Brazilian study that analyzed the use of persuasive food advertising strategies on three popular open television channels in the country showed a direct association between advertising aimed at children and ads showing high-calorie foods rich in fat, sugar, and sodium, such as sugary drinks, fast food, and sweet cookies [6].

Thus, it is worth mentioning that the consumption of UPFs has its negative effects enhanced in the context of screen exposure. Together, they can generate more distractions and imbalances in the hormonal regulation of hunger and satiety [31], which allied to the sedentary lifestyle can contribute to childhood overweight.

Among the limitations of this study, it is noteworthy that the content watched by the child during screen exposure was not investigated, and it was not determined what foods were consumed by the children during this exposure. The literature already shows that the educational use of screens can be positive for children [32]. However, there are controversies about the use of digital media during this life cycle, requiring more research for substantial conclusions. Furthermore, the child’s daily active time was not explored, since it could have influenced the results. The practice of physical exercise is related to favorable results in health indicators [20], such as the development of motor skills and the reduction in adiposity. The alteration or suspicions of developmental delays and/or behavioral problems was not considered an exclusion criterion, which may have introduced bias in the results; however, this variable was not correlated with the duration of screen exposure in children. Finally, the high number of missing data in the anthropometric measurements may have negatively influenced the statistical analyses, preventing findings of an association between this variable and the outcome.

Regarding this study’s potential, it should be noted that the authors are unaware of any international and national studies that, after the COVID-19 pandemic, investigated screen time in children aged 4 years and its association with child development through the screening instrument SWYC, and determined children’s eating habits through their UPF score. Thus, it is suggested that future studies carry out a long-term follow-up to understand the implications of screen exposure associated with UPFs consumption, in addition to investigating the cause-and-effect relationship between screen time and child development, with the application of diagnostic instruments. The comparison with parent–child data can also be encouraged.

## 5. Conclusions

Our findings suggest that high screen time by 4-year-olds is related to lower child development scores, an increased habit of eating in front of screens, and higher UPF scores. Therefore, it is necessary to create public policies that promote the greater control of screen exposure in that population. This can be achieved through the development of actions to raise awareness aimed at parents, caregivers, and educators, focusing on elucidating this problem, and warning about the harmful effects of excessive screen time on children.

## Figures and Tables

**Table 1 ijerph-21-01504-t001:** Characteristics of the main caregiver and the child according to screen exposure time, 2022–2023 (*n* = 362).

Variables	Sample	Screen Time (min)	*p*-Value
	*n* (%)/ Median (IQR)	Median (IQR)	
Caregiver’s characteristics			
Age (years)	32.5 (10)	-	r = −0.04; *p* = 0.42 ***
Caregiver			
Mother	346 (95.6)	120 (120)	0.12 *
Others	16 (4.4)	60 (90)	
Color			
White	69 (19.6)	120 (135)	0.30 **
Brown	174 (49.4)	120 (120)	
Black	90 (25.6)	120 (120)	
Asian	17 (4.8)	120 (150)	
Indigenous	2 (0.6)	240 (-)	
Material Status			
With a partner	243 (68.5)	120 (120)	0.84 *
Without a partner	112 (31.5)	120 (180)	
Level of education			
Finished elementary school	31 (8.6)	120 (130)	0.59 **
Finished high school	261 (72.1)	120 (120)	
College degree or above	70 (19.3)	120 (90)	
Professional occupation			
Paid occupation	255 (70.4)	120 (120)	0.55 *
Unpaid occupation	107 (29.6)	120 (120)	
Social benefits			
Yes	127 (35.1)	120 (180)	0.05 *
No	235 (64.9)	120 (120)	
Number of family members	4 (1)	-	r = −0.02; *p* = 0.68 ***
Total income	2500 (2500)	-	r = −0.08; *p* = 0.14 ***
Per capita income (Brazilian real)	666.67 (767)	-	r = −0.09; *p* = 0.01 ***
Child’s characteristics			
Age (months)	48 (0)	-	r = 0.04; *p* = 0.42 ***
Sex			
Female	176 (48.6)	120 (120)	0.92 *
Male	186 (51.4)	120 (120)	
Color			
White	122 (34)	120 (173)	0.52 **
Brown	173 (48.2)	120 (120)	
Black	51 (14.2)	120 (120)	
Yellow	8 (2.2)	120 (105)	
Indigenous	5 (1.4)	120 (185)	
Education			
Attends school or childcare center	321 (88.7)	120 (120)	0.01 *
Does not attend school or childcare center	41 (11.3)	180 (150)	
School time			
Full-time	147 (45.9)	120 (120)	0.06 *
Part-time	173 (54.1)	120 (143)	
Age that started school (months)	32 (15)	-	r = 0.02; *p* = 0.73 ***
Breastfeeding (months)	15 (20)	-	r = 0.01; *p* = 0.84 ***
Anthropometric measurements Weight-for-age			
Adequate	280 (92.7)	120 (120)	0.09 *
Inadequate	22 (7.3)	150 (120)	
Length-for-age			
Adequate	213 (96.8)	120 (120)	0.55 *
Inadequate	7 (3.2)	120 (150)	
BMI-for-age			
Underweight	14 (6.4)	120 (75)	0.56 **
Normal weight	151 (68.6)	120 (120)	
Overweight	55 (25)	120 (180)	
Alteration or suspicion of developmental delay and/or behavioral problems			
Yes	70 (19.8)	120 (169)	0.15 *
No	283 (80.2)	120 (120)	
Child development (score)	14 (5)	-	r = −0.15; *p* = 0.01 ***
Daily reading (days)	1 (3)	-	r = −0.16; *p* < 0.001 ***
Sleep time (hours)	10 (2)	-	r = −0.04; *p* = 0.44 ***
Eating in front of screens			
Yes	232 (64.3)	120 (105)	<0.001 *
No	129 (35.7)	90 (60)	
UPF score (points)	4 (3)	-	r = 0.12 *p* = 0.02 ***
Infection with SARS-CoV-2			
Yes	93 (26.1)	120 (120)	0.49 *
No	263 (73.9)	120 (120)	

* Mann–Whitney test; ** Kruskal–Wallis test; *** Spearman’s correlation. UPF: ultra-processed food; IQR: interquartile range; BMI: body mass index.

**Table 2 ijerph-21-01504-t002:** Generalized linear model of the factors associated with screen exposure, 2022–2023 (*n* = 362).

Variables	B	CI 95%	F-Test	*p*-Value
Child development (score)	−0.03	−0.04–0.01	7.54	0.01
Eating in front of screens ^a^	0.34	0.18–0.5	17.31	<0.001
UPF score (points)	0.05	0.02–0.09	9.11	0.001

R^2^ = 35.02. Backward method. F-test: *p* < 0.001. ^a^ No eating in front of screens as a reference. Adjusted for age, marital status, level of education and per capita income, and child’s infection with SARS-CoV-2. UPF: ultra-processed food; CI: confidence interval.

## Data Availability

The datasets generated and analyzed during the current study are not publicly available due to confidentiality issues but are available from the principal investigator upon reasonable request.

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
