# Peer review of "Screen Exposure in 4-Year-Old Children: Association with Development, Daily Habits, and Ultra-Processed Food Consumption"

_ijerph, 2024, doi:10.3390/ijerph21111504_

Round 1
Reviewer 1 Report
Comments and Suggestions for Authors
This manuscript is a modest study, although important for the research topic of the relationship between screen time, child development, and ultra-processed food choices.
It requires consideration:
- - please consider the title in the context of eating habits, not food consumption (ultra-processed food choices), as eating habits with an assessment of ultra-processed food choices were studied (correction throughout the text); the title also does not include researched relationships with parents
- - in keywords please consider 3 screen entries
- - the introduction lacks an explanation of why the accuracy of parent-child BMI in body mass classification was not examined
- - the introduction lacks a research hypothesis
- - the methodology does not include exclusion criteria for the child
- - in the methodology, the lack of questionnaire validation should be emphasized (or in the weak points of the study in the discussion)
- - it should be added in the description of the methodology how the group was selected, whether the research is representative
- - some data quoted in the methodology are not included in Table 1 (e.g. number of family members, etc.) - requires detailed correction
- - the consistency of the description of the statistical analysis with the results of Table 1 should be checked; Table 1 does not specify what the correlations refer to; Table 1 does not provide a description of abbreviations, e.g. UPF; Table 1 does not include the age of the children
- - the discussion lacks verification of the hypothesis, as well as the discussion lacks the strengths and weaknesses of the study - it requires supplementation
- - all cited items should be in square brackets
- - in the work, the comparison of parent-child data is poorly legible
Author Response
This manuscript is a modest study, although important for the research topic of the relationship between screen time, child development, and ultra-processed food choices.
Answer: We appreciate your attention when considering our work. All points placed were carefully reviewed and we hope to meet the journal’s expectations. In addition, a rereading and adaptation of all topics requested by the reviewers was carried out. All changes made were highlighted in the text.
Suggestion 1: please consider the title in the context of eating habits, not food consumption (ultra-processed food choices), as eating habits with an assessment of ultra-processed food choices were studied (correction throughout the text); the title also does not include researched relationships with parents
Answer: All text was rereading and the change was made as suggested. As the relationship with parents is not our research objective nor an important explanatory variable, we consider that this information should not be included in the title. However, the title was modified to facilitate broad understanding.
Suggestion 2: in keywords please consider 3 screen entries
Answer: We added “Daily screen time” to keywords.
Suggestion 3: the introduction lacks an explanation of why the accuracy of parent-child BMI in body mass classification was not examined
Answer: We appreciate the suggestion, but after reviewing the manuscript we believe that it is not appropriate to add this information to the introduction. The main objective of the study was to investigate the association between screen exposure and child development, eating habits, especially ultra-processed foods, and daily habits with 4-year-old children. Thus, despite the relevance of the suggested theme (the accuracy of parent-child BMI in body mass classification), it was decided not to make changes to direct the focus.
Suggestion 4: The introduction lacks a research hypothesis
Answer: The changes were made at the end of the fifth paragraph of the introduction (line 55-57).
Suggestion 5: the methodology does not include exclusion criteria for the child
Answer: We reread and rewrote the excerpt in the last paragraph of topic 2.1: "and those families that did not answered about daily screen time (n=70) were excluded, totaling 362 caregiver-child pairs". No other exclusions in the sample were performed.
Suggestion 6: in the methodology, the lack of questionnaire validation should be emphasized (or in the weak points of the study in the discussion)
Answer: A questionnaire to be applied to families was developed and contained sociodemographic questions (age, sex, marital status, for example), questions about the child's daily habits (screen time, sleep time, for example) and complete questionnaires validated for the Brazilian population (such as the SWYC and NOVA of ultra-processed foods score). We reread and rewrote the excerpt in topic 2.2 and we added information about the validated questionnaires (lines 121-122 and lines 138-140).
Suggestion 7: it should be added in the description of the methodology how the group was selected, whether the research is representative
Answer: We agree with your point of view, and to improve clarity of the text, we have inserted this information in the last two paragraphs of the topic "2.1. Study design, sample, and data collection". This change will ensure that the group selection methodology and the representativeness of the research are explicitly addressed in the context of the study design (lines 83-91).
Suggestion 8: some data quoted in the methodology are not included in Table 1 (e.g. number of family members, etc.) - requires detailed correction
Answer: The suggestion was made and information about the caregiver, the number of residents in the residence, the total income and the child's age (in months) were entered in Table 1.
Suggestion 9: the consistency of the description of the statistical analysis with the results of Table 1 should be checked; Table 1 does not specify what the correlations refer to; Table 1 does not provide a description of abbreviations, e.g. UPF; Table 1 does not include the age of the children
Answer: The requested changes were made as described below:
- Consistency of the description of the statistical analysis with the results in Table 1: We revised the description of the statistical analysis to ensure that it is consistent with the results presented in Table 1.
- Correlation of variables in Table 1: We clarified what the correlations refer to (Spearman's correlation) by adding this information in the footer of Table 1. (line 190)
- Description of abbreviations: We included an explanation for the abbreviation "UPF" (and other abbreviations) in the footer of Table 1. (line 191)
- Age of children in Table 1: We inserted information about the age of the children in Table 1.
Suggestion 10: the discussion lacks verification of the hypothesis, as well as the discussion lacks the strengths and weaknesses of the study - it requires supplementation
Answer: We appreciate your suggestion to improve the discussion including hypothesis verification, as well as the strengths and weaknesses of the study. We believe that these aspects are crucial for a more robust analysis of the results. The first paragraph of the discussion was revised to address the suggestions regarding the hypothesis (lines 208-210) and the last two paragraphs of the discussion were reviewed and adjusted as suggested.
Suggestion 11: all cited items should be in square brackets
Answer: The change was made as suggested.
Suggestion 12: in the work, the comparison of parent-child data is poorly legible
Answer: This type of comparison was not carried out since the main focus of the study was on the child, so much so that any primary caregiver (mother, father, grandmother) could answer the questionnaire. As requested in another part of the opinion, other general information was added - such as information about the caregiver, the number of residents in the residence and the total income. We add the investigation these data as a possibility for future studies (lines 293-294).
Reviewer 2 Report
Comments and Suggestions for Authors
Dear Authors,
The paper deals with very important aspects of public health - it links selected sedentary behavior, daily hygiene practise and consumption of processed foods in a group of very young children. However, I have trouble determining what type of study it was - once the authors indicate a prospective study (clinical-control, cohort?) and in the abstract there is a cross-sectional study (which is not prospective in my opinion). The discussion promises to be interesting - it could be expanded especially to include processed food issues.
Additional remarks:
- The title as well as the purpose of the study should be changed - in the publication, processed food consumption was analyzed and this should be highlighted as well as selected sleep hygiene behaviors, mainly
- Please specify in the methodology the type of study?
- Did the exclusion criteria apply only to women? Did only female caregivers participate in the study? Please elaborate. Please provide the inclusion criteria for the study.
- Was it questioned about the number of family members cohabiting the household or the number of children?
- Line 109: Is this questionnaire suitable for children? was it validated for use in children population?
- Table 1 - The title of the table should be changed, it contains not only the characteristics of the studied group
- Line 182-183 - Please rewrite
The major limitation of this study or the presentation of the results is the absence of nutritional status of the subjects. The foods we eat affect our nutritional status as well as identify different dietary patterns in different categories of nutritional status assessment. This information for both caregivers and children should be included in the publication. hors themselves once point to a prospective study and the abstract reports a cross-sectional study.
Author Response
Dear Authors,
The paper deals with very important aspects of public health - it links selected sedentary behavior, daily hygiene practise and consumption of processed foods in a group of very young children. However, I have trouble determining what type of study it was - once the authors indicate a prospective study (clinical-control, cohort?) and in the abstract there is a cross-sectional study (which is not prospective in my opinion). The discussion promises to be interesting - it could be expanded especially to include processed food issues.
Answer: We appreciate your attention when considering our work. All points placed were carefully reviewed and we hope to meet the journal’s expectations. In addition, a rereading and adaptation of all topics requested by the reviewers was carried out. All changes made were highlighted in the text.
Additional remarks:
Suggestion 1: The title as well as the purpose of the study should be changed - in the publication, processed food consumption was analyzed and this should be highlighted as well as selected sleep hygiene behaviors, mainly
Answer: The objective and title were rewritten as suggested.
“Therefore, the aim of this study was to investigate the association between screen exposure and child development, eating habits, especially ultra-processed foods, and daily habits with 4-year-old children.” (lines 58-60)
Suggestion 2: Please specify in the methodology the type of study?
Answer: We added this information on first line of topic 2.1 (line 64)
Suggestion 3: Did the exclusion criteria apply only to women? Did only female caregivers participate in the study? Please elaborate. Please provide the inclusion criteria for the study.
Answer: The exclusion criteria mentioned applied to postpartum women in the baseline cohort conducted in 2018 and 2019. In this cohort, all participants were women who were at postpartum with their babies.
In the current investigation, the focus was on the children of women who participated in the previous study, in order to continue the follow-up.
As for the inclusion criteria, we contacted all mothers to participate in the study, however, the questionnaire could be answered by them or by another main caregiver (information added in Table 1).
About the exclusion criteria, the questionnaires whose caregiver did not provide information about screen time were excluded (n=70). This information was added in the last paragraph of topic 2.1 of “Materials and Methods”.
Suggestion 4: Was it questioned about the number of family members cohabiting the household or the number of children?
Answer: We added this information on Table 1.
Suggestion 5: Line 109: Is this questionnaire suitable for children? was it validated for use in children population?
Answer: We appreciate the question raised about the suitability of the questionnaire for the child population and its validation for this group. The questionnaire was originally validated for the Brazilian adult population. The choice to apply this instrument to the child population in the present study is based on the scarcity of similar instruments validated for this age group, making the questionnaire a relevant option. Additionally, children's eating habits tend to reflect those of their parents or caregivers, which strengthens the justification for using this instrument, given the familial influence on dietary habits. Furthermore, a recent Brazilian study applied the same instrument with school-aged children (6-11 years), as published in PLOS ONE (https://journals.plos.org/plosone/article?id=10.1371/journal.pone.0294871#abstract0). Although our study includes a younger age group, we believe that this previous application in school-aged children supports the feasibility of the instrument for use in pediatric populations, especially when used in conjunction with the family context and parental data on food consumption. Thus, we believe that the application of the questionnaire to our study group is justified, considering the scarcity of instruments for this age group, family dynamics, and the precedent for using the instrument in studies with children.
Suggestion 6: Table 1 - The title of the table should be changed, it contains not only the characteristics of the studied group
Answer: We appreciate the suggestion to change the title of Table 1. The title "Table 1 - Characteristics of the main caregiver and the child according to screen exposure time, 2022-2023 (n=362)" was crafted to accurately represent the data described, including the characteristics of the main caregiver and the children participating in the study according to screen exposure time. Thus, we understand that the title aligns with the content of the table and provides a clear and straightforward description of the information presented.
Suggestion 7: Line 182-183 - Please rewrite
Answer: The lines were rewritten as suggested.
Suggestion 8: The major limitation of this study or the presentation of the results is the absence of nutritional status of the subjects. The foods we eat affect our nutritional status as well as identify different dietary patterns in different categories of nutritional status assessment. This information for both caregivers and children should be included in the publication. hors themselves once point to a prospective study and the abstract reports a cross-sectional study.
Answer: We agree with your point of view and included the child's nutritional status data in Table 1. Since this information must be measured by a health professional in person, we had a high number of missing data (missing height data, n=142). However, we agree on the importance of this variable for the outcome studied, and we included it as a limitation (lines 281-284), while we suggest the investigation of these data as a possibility for future studies. (lines 293-294).
Round 2
Reviewer 1 Report
Comments and Suggestions for Authors
I accept the text corrections; the Authors have largely taken into account the reviewer's suggestions.
Of the 3 definitions in the keywords "Screen exposure; Screen time; Daily screen time" I suggest leaving one.
Reviewer 2 Report
Comments and Suggestions for Authors
Thank you for considering the proposed changes and clarifying ambiguities. As it currently is, I have no comments on the manuscript.